# Fibrosis in Chronic Kidney Disease: Pathophysiology and Therapeutic Targets

**DOI:** 10.3390/jcm13071881

**Published:** 2024-03-25

**Authors:** Allison B. Reiss, Berlin Jacob, Aarij Zubair, Ankita Srivastava, Maryann Johnson, Joshua De Leon

**Affiliations:** Department of Medicine and Biomedical Research Institute, NYU Grossman Long Island School of Medicine, Mineola, NY 11501, USA; bjacob05@nyit.edu (B.J.); azubair@student.touro.edu (A.Z.); ankita.srivastava@nyulangone.org (A.S.); maryann.johnson@nyulangone.org (M.J.); joshua.deleon@nyulangone.org (J.D.L.)

**Keywords:** chronic kidney disease, inflammation, nuclear factor kappa B, renal fibrosis, SGLT2 inhibitors

## Abstract

Chronic kidney disease (CKD) is a slowly progressive condition characterized by decreased kidney function, tubular injury, oxidative stress, and inflammation. CKD is a leading global health burden that is asymptomatic in early stages but can ultimately cause kidney failure. Its etiology is complex and involves dysregulated signaling pathways that lead to fibrosis. Transforming growth factor (TGF)-β is a central mediator in promoting transdifferentiation of polarized renal tubular epithelial cells into mesenchymal cells, resulting in irreversible kidney injury. While current therapies are limited, the search for more effective diagnostic and treatment modalities is intensive. Although biopsy with histology is the most accurate method of diagnosis and staging, imaging techniques such as diffusion-weighted magnetic resonance imaging and shear wave elastography ultrasound are less invasive ways to stage fibrosis. Current therapies such as renin-angiotensin blockers, mineralocorticoid receptor antagonists, and sodium/glucose cotransporter 2 inhibitors aim to delay progression. Newer antifibrotic agents that suppress the downstream inflammatory mediators involved in the fibrotic process are in clinical trials, and potential therapeutic targets that interfere with TGF-β signaling are being explored. Small interfering RNAs and stem cell-based therapeutics are also being evaluated. Further research and clinical studies are necessary in order to avoid dialysis and kidney transplantation.

## 1. Introduction

Chronic kidney disease (CKD) is a progressive, irreversible condition characterized by reduced kidney function [1,2]. It is a major risk factor for end-stage renal disease (ESRD) and cardiovascular disease and a leading cause of mortality worldwide [3,4,5]. CKD is defined by a structural or functional change in the kidney or a glomerular filtration rate (GFR) of less than 60 mL/min/1.73 m^2^, or both for a duration of at least 3 months [6,7]. The condition can be staged or classified according to albuminuria, GFR, and cause [8]. In the Western world, diabetic nephropathy and hypertensive renal damage are two of the most prevalent root causes of CKD [9]. The United States Centers for Disease Control and Prevention (CDC) reports that 1 in 3 adults with diabetes and 1 in 5 adults with hypertension may have CKD [10]. CKD is asymptomatic in the early stages and, therefore, may go undiagnosed for extended periods. Approximately 37 million people in the United States meet the criteria for CKD, and many are unaware that they have the condition. Although there is no cure, treatment aims to preserve kidney function and prevent progression by adequately controlling blood pressure, glucose levels, proteinuria, and metabolic acidosis through dietary/lifestyle changes and pharmacological interventions [11]. In other areas of the world, such as in Asia and Africa, the leading reasons for developing CKD are glomerulonephritis and CKD of unknown origin. Diseases such as HIV and the nephrotoxic treatments associated with the disease can cause direct glomerular interstitial damage, contributing significantly to the development of CKD. CKD-related deaths have been increasing around the world, and CKD is on track to become the fifth most common cause of death globally by the year 2040. Late diagnosis and limited therapeutic tools are two factors that are influencing these dim predictions [1].

Kidney fibrosis is the predominant pathophysiologic change observed with progression of CKD [12,13]. Fibrosis occurs when there is an excessive accumulation of extracellular matrix (ECM) within the renal parenchyma. Excess fibrotic deposition alters the normal kidney architecture, impeding normal blood supply and consequently leading to irreversible kidney injury [14,15].

This review is focused on multiple aspects of the fibrotic process in CKD. The major areas covered include the clinical evaluation, biomarkers, and imaging modalities that can aid in diagnosing and monitoring progression of fibrosis as well as the potential therapeutic agents that can hinder its progression and preserve renal function.

## 2. Fibrosis in Chronic Kidney Disease: Overview

Fibrosis impacts various kidney compartments beginning in the tubulointerstitial area and eventually affecting the vasculature and glomeruli, thus contributing to the development of arteriosclerosis and glomerulosclerosis [16,17,18,19]. Fibrotic burden is a key indicator of future adverse renal outcomes, for which there are no safe and effective therapies. Fibrotic progression occurs silently and often without any initial overt presentation [20]. Increases in blood pressure, serum creatinine, or urinary albumin excretion manifest only when the majority of the kidney becomes fibrotic. Assessment of the degree of fibrosis during the initial silent phase can be crucial for the identification of patients who are at high risk of CKD progression before serious renal dysfunction has developed [20,21].

As the organ undergoes fibrotic changes, progressive capillary loss occurs concurrently [22]. Reduced perfusion of the renal parenchyma results in decreased oxygen delivery to tubular epithelial cells, promoting apoptosis, inflammation, and release of profibrotic stimuli that worsen fibrogenesis [23,24,25]. Similarly, another feature of the fibrosing kidney is increased renal stiffness [26]. Organ stiffening occurs via the substitution of compliant cells with a rigid matrix through the crosslinking of matrix fibrils [20,27,28]. In a rat model, fibroblasts are unresponsive when activated in a soft healthy organ environment but respond to transforming growth factor (TGF)-β, a profibrotic stimulus, when in the presence of a stiff environment, as seen in injured or fibrotic liver [29]. Renal stiffening and capillary dropout are not only manifestations of fibrosis but also key contributors to its progression [30].

Effective detection methods can aid in guiding clinical treatment plans and assessing the efficacy of new antifibrotic agents. Differentiating and quantifying irreversible scar burden from reversible injury induced by inflammation is vital when predicting the amount of renal function recovery post-treatment [31].

## 3. Clinical Assessment of Kidney Fibrosis

### 3.1. Kidney Biopsy

Currently, there is no optimal assessment tool that can monitor kidney fibrosis in the clinical setting. Kidney biopsies are considered the gold standard in the assessment of renal fibrosis [32,33]. Biopsy is usually performed percutaneously with real-time ultrasound guidance, but laparoscopic or open routes are also possible [34]. There can be several drawbacks to this invasive method that include sampling bias, observer variability, and complications resulting from tissue sampling [35]. Biopsies are prone to significant bleeding risk and consist of sections that are normally 2 mm in diameter [36,37,38]. Since fibrosis is focal and distributed heterogeneously and biopsy samples are only representative of 1% of the kidney, sampling bias could affect results [39]. Due to these safety and sampling issues, many patients do not undergo a biopsy, and fibrotic burden is not determined or is not followed over time in this way [40,41,42]. Without this key piece of knowledge, clinicians are left without this valuable insight into the underlying renal pathology when determining appropriate medical management. They then must rely on surrogate measurements of kidney function such as estimated GFR [43].

### 3.2. Imaging

Diagnostic imaging techniques such as magnetic resonance imaging (MRI) and ultrasound are noninvasive alternatives to biopsy for the assessment of kidney fibrosis [44,45,46,47]. Both ultrasound and MRI techniques can indirectly quantify renal fibrosis by assessing functional, mechanical, and molecular changes in the organ due to vascular obstruction, tubular atrophy, and kidney shrinkage secondary to fibrotic changes [42,48,49]. The advantages of conventional ultrasound are the low expense and avoidance of ionizing radiation exposure, but differentiating fibrosis from inflammation based on increased parenchymal echogenicity is not reliable [50,51,52]. Shear wave elastography (SWE), an ultrasound technique that objectively measures tissue stiffness, is a better option than standard ultrasound because it can detect increasing tissue stiffness and loss of elasticity with fibrosis [53]. This will be discussed in more detail in a later subsection.

In murine models of CKD, MRI has shown promise in estimating fibrosis. While in humans the correlation between renal function and MRI evaluation has not always been clearcut, technology is improving rapidly [54,55].

Diffusion-weighted MRI (DWI) does not require contrast media and is therefore less hazardous to compromised kidneys [56]. It is able to measure water movement in the tissues of the kidney and capillary loss. DWI and SWE imaging techniques are discussed in the subsections that follow.

#### 3.2.1. Diffusion-Weighted Magnetic Resonance Imaging

DWI can be utilized to estimate the overall integrity of the kidney and microvascular blood flow. It has the potential to gauge changes in tissue microstructure indicating the presence and degree of fibrosis by evaluating whether water diffusion is restricted due to tubular atrophy and extracellular matrix accumulation [57,58]. The technique employs magnetic field gradients to measure the movement of water molecules. This noncontrast method images both directional water motion (i.e., blood and urine flow) and random intra/extracellular water motion using diffusion weighting, which is based on random Brownian motion of molecules in intracellular and extracellular settings [59]. A quantitative parameter that integrates both the random diffusion and directional movement of water is the apparent diffusion coefficient (ADC). The ADC provides insight into the extent of water diffusion in extracellular/extravascular spaces and capillary perfusion [31,60]. Both preclinical and clinical studies have demonstrated lower renal ADC values in dysfunctional renal transplants and native renal pathologies. The utility of ADC in assessing renal fibrosis was tested by Togao using a mouse model of unilateral ureteral obstruction [61]. Decreased renal cortical ADC values correlated with increased cellular density and α-smooth muscle actin expression. Clinical studies have also demonstrated a correlation between renal ADC values and histopathological fibrotic scoring. Low ADC values can be indicative of decreased water diffusion due to potential changes in the microstructure from tubular atrophy or interstitial fibrosis [62,63]. However, there are still limitations to this method since various other factors can affect water mobility such as intravascular volume, diuretic use, edema, and urine flow rate [64,65].

Multifunctional MRI gives more detailed information by using combinations of techniques such as perfusion, diffusion and blood oxygen level-dependent (BOLD) imaging [66]. It is not in common clinical use, but Mao et al. recently showed the power of this modality in evaluating interstitial kidney fibrosis [67].

#### 3.2.2. Shear Wave Elastography

SWE utilizes ultrasound-generated shear wave velocity to estimate renal stiffness. The technique uses focused acoustic energy waves to induce tissue displacement microscopically [68,69]. The resulting tissue displacement creates perpendicular shear waves that are followed sonographically as they move through the tissue. Stiffer tissues correlate with higher shear wave velocity [70,71]. This technique has been approved by the FDA to aid in cirrhotic liver detection and has been shown to have a high sensitivity and specificity [72,73]. However, there have been conflicting reports about the effectiveness of SWE in measuring renal fibrosis [74]. This is due, in part, to several factors that can influence intrarenal stiffness besides fibrosis. These include tissue perfusion, tubular or interstitial pressure, and tissue anisotropy. Accumulating evidence demonstrates that decreased renal perfusion can potentially conceal the presence of renal fibrosis by reducing SWE results [31,75].

#### 3.2.3. Biomarkers of Kidney Fibrosis

A urine specimen is easy to collect, and urine composition can yield much useful information about kidney function. These factors make urine an excellent source of measurable biomarkers that can aid in the detection of renal fibrosis [76]. Reliable biomarkers such as increased urine TGF-β1 and connective tissue growth factor (CTGF) levels have been seen in individuals with progressive renal diseases and can serve as indicators of renal fibrosis [77,78]. Increased urinary excretion of type IV collagen is associated with glomerular ECM accumulation, and this has been demonstrated in animal studies and in patients with IgA nephropathy and diabetic kidney nephropathy with deteriorating renal capability [79,80]. Urinary and serum markers of collagen type III turnover have been associated with CDK progression and fibrosis [80,81,82].

A study conducted by Melchinger et al. investigated the relationship between urine uromodulin, the most abundant protein found in urine, and kidney histological findings in both humans and mice [83]. Uromodulin is a glycoprotein that serves as an indicator of kidney tubular health and is produced in the ascending limb of the loop of Henle and early distal convoluted tubule [84]. The study demonstrated that higher urinary uromodulin levels correlated independently with lower tubulointerstitial fibrosis in patients enrolled in the Yale biopsy cohort undergoing a kidney biopsy and also in a mouse model of kidney fibrosis induced through unilateral ischemia/reperfusion injury to the left kidney.

A systematic review from Mansour and colleagues found that three specific biomarkers, matrix metalloproteinase (MMP)-2, TGF-β, and monocyte chemoattractant protein (MCP)-1, were independently associated with worsening renal function. Of these three, TGF-β exhibited the strongest association with fibrosis on biopsy and correlated with worse renal outcomes [85,86]. This association is further validated in mouse studies demonstrating that the overexpression of TGF-β by renal tubular epithelial cells promotes tubulointerstitial fibrosis and that fibrotic progression can be reduced through the blocking of this growth factor [87,88,89]. MCP-1 receptor blockade in preclinical studies was associated with reduced interstitial fibrosis and serves as a potential biomarker with a strong association to the advancement of renal disease [90,91]. MMP-2, a gelatinase produced by glomerular and tubular cells, contributes to epithelial–mesenchymal transition, a critical process in kidney fibrosis [92,93]. Kidney fibrosis is attenuated under conditions of unilateral ureteral obstruction in mice with reduced MMP-2 production due to a deficiency in the MMP-2 inducer Basigin, and MMP-2 knockout in mice also results in reduced kidney fibrosis under conditions of unilateral ureteral obstruction [94,95]. MMP-2 serves as a strong independent predictor of deteriorating estimated GFR [96].

## 4. Pathophysiology of Fibrosis in the Setting of Chronic Kidney Disease

### 4.1. The Fibrotic Niche and the Profibrotic Microenvironment

Fibrotic lesions are neither uniform nor random within the renal parenchyma and are often interspersed between areas of normal kidney tissue [97]. Observation of these characteristics led to the concept of the fibrogenic niche, a specialized tissue microenvironment composed of the injured parenchyma and multiple non-parenchymal cell lineages that contribute to the focal activation of fibroblasts in discrete locations [98,99]. This fibrotic niche, whose structural constituents include kidney resident cells, mesenchymal cells, infiltrating immune cells, ECM network, extracellular vesicles, soluble factors, and metabolites, fosters fibroblast activation and ECM deposition and expansion. The cellular aspects such as the resident and infiltrating cells are motile elements of the niche, while the extracellular vesicles, secreted factors, and metabolites serve as diffusional components. The ECM network plays a vital role as a stationary anchor for cellular adhesion within the fibrotic niche [100]. ECM proteins are categorized as structural (collagens, fibronectins, and elastin), matricellular (fibrillin-1, tenascin-C, CTGF, periostin), matrix-modifying proteins, and proteoglycans [99]. Matricellular proteins are the most prevalent within the fibrotic kidney and can influence various cellular processes such as migration, apoptosis, ECM assembly, inflammation, wound healing, and fibrosis. These proteins serve as signal reservoirs and can aggregate growth factors and cytokines from the extracellular environment. Moreover, they influence cell behavior and serve as signal presenters by aiding the binding of extracellular matrix-associated ligands to their respective plasma membrane receptors. [97]. Specifically, the matricellular protein tenascin-C is a key contributor in the formation of the fibrotic niche. It promotes the further activation and proliferation of fibroblasts as well as the production and deposition of ECM through the activation of the integrin/adhesive plaque kinase (FAK)/mitogen-activated protein kinase (MAPK) signaling pathway [101]. Moreover, this hexametric protein can attract and confine profibrotic mediators such as Wnts, hedgehog, and TGF-β, fostering a microenvironment conducive to accelerating the fibrotic process. CTGF is a key mediator in the development of tubulointerstitial fibrosis in CKD. The protein serves as a downstream mediator of TGF-β1 and regulates the expression of proteins such as bone morphogenetic protein and interacts with various other soluble factors, ligands, and receptors that modulate cell proliferation, migration, differentiation, and the progression of fibrosis [102]. Fibrillin-1 is a matricellular glycoprotein that is also a major component of the fibrotic niche. The protein encourages apoptosis and prevents the proliferation of vascular endothelial cells [103]. Disrupting or inhibiting key players involved in fibroti niche formation are already being explored for future CKD therapy [104].

### 4.2. Cellular Mechanisms—Dysregulation of TGF-B/Smad in Promoting Renal Fibrosis

The classical TGF-β (transforming growth factor-β) signaling pathway is a key mediator in the promotion of renal fibrosis [105,106]. Renal fibrosis, the massive deposition of extracellular matrix and loss of renal function characterized by the presence of renal myofibroblasts, is particularly concerning as it is an irreversible process. Interstitial myofibroblasts are cells that produce an abundance of α-smooth muscle actin and are central in wound repair. However, the constant activation of these cells drives renal fibrosis. These myofibroblasts, which amplify the renal fibrogenic cascade, are directly linked to TGF-β [107]. TGF-β can induce its fibrotic effects through multiple mechanisms. However, the main fibrosis-generating pathway is through TGF-β1/Smad (an acronym coined by the fusion of Caenorhabditis elegans Sma genes and the Drosophila Mad, Mothers against decapentaplegic) [108,109,110].

In this canonical signaling pathway, TGF-β1 is secreted in latent form; once activated, mature TGF-β1 binds to the TGF-β receptor (TGFBR)2 on the cell surface, which then phosphorylates and activates TGFBR1 (also known as ALK5). TGFBR1 then phosphorylates Smad2 and Smad3 at their C-terminal ends. Smad2 and Smad3 then form a complex with Smad4 that translocates to the nucleus [111,112]. The Smad3 component of the complex can bind directly to gene promoters to induce transcription of downstream profibrotic molecules such as fibronectin, α-smooth muscle actin, and collagen. The accumulation of fibronectin and α-actin in the glomerular and tubulointerstitial portion of the kidneys triggers glomerulosclerosis and tubulointerstitial fibrosis, respectively. While TGFBR1 and TGFBR2 act as serine/threonine kinases in this pathway, they can also function as tyrosine kinases and act downstream through the MAPK–extracellular signal-regulated kinase (ERK) pathways [113,114].

Through this pathway, TGF-β1/Smad are able to trigger glomerulosclerosis, interstitial fibrosis, and inflammation. This scarring coupled with the anti-inflammatory effects of TGF-β1 enhances the progression of CKD to ESRD [115,116]. Additionally, under pathological conditions, the expression of the intracellular proteins Smad2 and Smad3 are upregulated while Smad7 expression is downregulated [115]. Targeting the TGF-β1/Smad pathway is considered the best route to effective therapy for prevention of renal fibrosis and progression to CKD/ESRD.

Mesangial cells also play an important role in the pathogenesis of CKD. These smooth muscle-like contractile cells contain actin and myosin and are the main component of the glomerulus, comprising roughly 30–40% of total glomerular cells [117]. In conjunction with endothelial cells and podocytes, mesangial cells form the glomerulus, which allows for ultrafiltration of blood plasma by size, shape and charge so that water and low-molecular-weight solutes can pass through while large plasma proteins are restricted from crossing the glomerular barrier [118]. Mesangial cells are the major producers of glomerular extracellular matrix (ECM), which provides structural support for glomerular capillaries. However, when activated in pathological conditions, mesangial cells proliferate, secrete several types of inflammatory cytokines, and synthesize an excess of matrix constituent proteins [117]. This leads to reduced glomerular function and contributes to the development of kidney fibrosis. TGF-β1 activates mesangial cells and, in a nephritic rat model, suppression of TGF-β1 expression decreases actin and matrix production [119,120]. Murine studies have provided insight into the role of mesangial cells in kidney fibrosis, especially in the diabetic state, where high glucose leads to oxidative stress and inflammation. Li et al. demonstrated that lysophosphatidic acid (LPA), a phospholipid which is elevated in diabetes and associated with carbohydrate-responsive element-binding protein (ChREBP), induces renal fibrosis in db/db mice through a mechanism that likely involves TGF-β1 [121]. These transgenic mice are an established model of obesity-induced type 2 diabetes, and they develop diabetic nephropathy with mesangial matrix accumulation [122]. In Li’s study, TGF-β1 levels were higher in the renal cortex in db/db mice than in wild-type mice, and LPA enhanced expression of TGF-β1 and fibronectin in mesangial cells from db/db mice. They found an association between upregulation of TGF-β1 and enhanced matrix accumulation and concluded that LPA drives mesangial matrix accumulation under diabetic conditions via TGF-β1. Additionally, Kim et al. displayed that by utilizing an LPAR1/3 antagonist, ki16425, there was inhibition of ChREBP expression and thus a decrease in LPA-induced fibrotic factors in these mice [123].

### 4.3. Diabetes

Diabetes is a major risk factor in the progression of CKD. In fact, type 2 diabetes is the most common cause of CKD worldwide [124]. The leucine-rich α-2-glycoprotein 1 (LRG1) is a proangiogenic factor that potentiates diabetic kidney disease [125,126]. Hong et al. demonstrated that the development of diabetic nephropathy, glomerular angiogenesis, and podocyte loss was decreased in streptozotocin-induced diabetic mice with LRG1 deficiency [127]. Additionally, LRG1 gene ablation was associated with reduced ALK1-Smad1/5/8 activation in the glomeruli of these diabetic mice. ALK1-Smad1/5/8 activation induces the formation of glomerular angiogenesis by inducing endothelial cell migration, proliferation, and tubular formation. The loss of LRG1 in mouse models was associated with decreased Smad3 phosphorylation in tubular epithelial cells and attenuated TGF-β1-induced tubulointerstitial fibrosis [128].

### 4.4. Epithelial–Mesenchymal Transition and the Macrophage

The biological process of epithelial–mesenchymal transition (EMT) constitutes a transdifferentiation of renal tubular epithelial cells (RTEC) in which they lose their characteristic features and undergo a structural change, taking on a more myofibroblastic phenotype [93,129]. With this change, epithelial-specific markers such as E-cadherin and cytokeratin are reduced, while the expressions of mesenchymal markers, such as α-SMA and desmin, are increased, and ECM overproduction ensues [130]. EMT is a key early phase of fibrosis that is driven largely by TGF-β and is potentially reversible [131,132] (Figure 1).

Macrophage infiltration and crosstalk between macrophages and transitioning RTEC can drive renal fibrosis. Macrophages induce RTEC to take on a mesenchymal phenotype [133,134,135,136]. Duffield et al. found that in a mouse model of crescentic glomerulonephritis, macrophage ablation led to decreased expansion of the renal myofibroblast population, reduced interstitial fibrosis, and protected renal function [137].

The polarization state of macrophages within the microenvironment can determine how they will affect RTEC [138]. Both classically activated M1 macrophages and alternatively activated M2 macrophages may contribute to EMT, but the exact role of each phenotype is undetermined [139,140]. Interestingly, a number of studies have found the M2 macrophage subtype, generally characterized as reparative and anti-inflammatory, to be important in EMT [134,141,142]. This results from an exaggeration of profibrotic processes driven by M2 macrophage participation in overzealous healing and repair [143].

One mechanism by which macrophages influence EMT is through the release of matrix metalloproteinases (MMPs) [144,145,146]. MMPs are calcium-dependent zinc-containing endopeptidases that are responsible for tissue remodeling and ECM degradation [147]. They function as anti-fibrotic agents, but when secreted by macrophages they can promote fibrosis through mechanisms such as EMT stimulation and TGF-β activation [148].

In particular, MMP-9 from murine macrophage-conditioned media induces EMT of cultured mouse tubular epithelial cells [149]. Further, MMP-9 knockout mice with renal fibrosis induced by double-ligation of the left ureter exhibit less fibrosis than wild-type mice subjected to the same ureter-obstructing procedure [150].

Additionally, there is evidence suggesting that exosomes can drive a positive feedback loop between renal parenchymal cells and interstitial macrophages that further drives the process of renal fibrosis. Lu et al. extracted exosomes from cultured RTEC that had been induced by TGF-β to undergo EMT. They then added these exosomes to the culture media of RAW264.7 murine macrophages and showed that the exosomes drove M1 polarization and changes in cell morphology [151]. Exosomes isolated from RTEC that had not been treated with TGF-β did not stimulate an increase in M1 macrophage-related markers. Moreover, when exosomes from RTEC that were undergoing EMT were injected into living mice, not only did the levels of M1 macrophage-related markers increase, but the markers of EMT and renal fibrosis were also elevated, creating a positive feedback effect that perpetuated EMT and renal fibrosis.

### 4.5. ICAM-1-Induced Epithelial–Mesenchymal Transition

Intercellular adhesion molecule (ICAM)-1, expressed by endothelial cells, lymphocytes and macrophages, plays an essential role in many inflammatory reactions, including renal tubular EMT [152]. Morishita et al. found that EMT induction of human-kidney (HK)-2 proximal tubular cells by TGF-β1 was accelerated by ICAM-1 in a model system where the HK-2 cells were co-cultured with human peripheral blood mononuclear cells (PBMC) [153]. ICAM-1 on the HK-2 cells interacted with lymphocyte function-associated antigen (LFA)-1 on the PBMC to activate the ERK1/2 pathway, and this resulted in more rapid EMT of HK-2 cells.

Renal ischemia induces chronic kidney damage with activation of renal tubular cell EMT [154,155]. Kelly et al. demonstrated that administration of antibodies specific to ICAM-1 to male Sprague-Dawley rats with bilateral renal ischemia protected against renal failure, quelling a potential inducer of renal EMT [156]. Consequently, studies focused on the downregulation of ICAM-1 in animal models and patients with CKD may be beneficial and are being studied further [157,158].

### 4.6. Serum Amyloid a and Chronic Kidney Disease

The acute-phase protein serum amyloid A (SAA) has been implicated in CKD and its circulating level inversely correlates with renal function [159,160]. It has value as a biomarker for CKD, as shown in murine studies [161]. In mice with renal fibrosis induced via unilateral ureteral obstruction, SAA was expressed at high levels in the obstructed kidney, and its depletion slowed fibrosis progression [162]. Feng et al. found that fatty acid-binding protein 4 (FABP4) mediated SAA upregulation in mice with unilateral ureteral obstruction and knockout; or pharmacologic inhibition of FAB4 prevented SAA-driven kidney fibrosis [163].

## 5. Nephroprotective Drugs in Current Use

Drugs in clinical use that are nephroprotective and can reduce renal fibrosis in both humans and experimental animal models of CKD include the renin-angiotensin blockers, mineralocorticoid receptor (MR) antagonists, and sodium/glucose cotransporter 2 inhibitors (SGLT2i). Each of these pharmacological agents will be discussed here.

### 5.1. Renin-Angiotensin Blockers

Treatment options that inhibit the renin-angiotensin-aldosterone system (RAAS) effectively reduce proteinuria and slow CKD progression and renal fibrosis. However, their benefit in early-stage CKD in persons without diabetes is unclear [164]. These medications reduce the activity of TGFβ, plasminogen activator inhibitor-1 (PAI1), and platelet-derived growth factor (PDGF), all molecules involved in the pathogenesis of renal fibrosis [12,165,166,167].

Angiotensin-converting enzyme inhibitors (ACEI) and angiotensin receptor blockers (ARBs) are first-line treatments shown to decrease kidney fibrosis via inhibition of the RAAS [168,169]. Hyperactivity of the RAAS worsens both systemic and glomerular capillary blood pressure, leading to kidney injury and renal fibrosis through the activation of proinflammatory and profibrotic pathways [170].

Binding of the octapeptide hormone angiotensin II to the angiotensin type 1 receptor (AT1R) promotes numerous profibrotic downstream signaling pathways in the kidney [171,172]. These effects are executed mainly through TGF-β, which acts in multiple ways. TGF-β promotes the synthesis of tissue inhibitors of metalloproteinases (TIMPs) that block the activity of MMPs. This favors imbalance between TIMP/MMP that encourages ECM production and the amplification of other profibrotic signaling pathways [173]. TGF-β also exerts pro-fibrotic effects via induction of a potent downstream mediator connective tissue growth factor (CTGF, also known as CCN2) [174,175,176]. CTGF is a cysteine-rich, 349-amino-acid matricellular protein that is upregulated in persons with kidney fibrosis [177,178]. CTGF expression is also induced by AngII [179]. CTGF promotes EMT and increases the production of ECM [180,181]. It works synergistically with TGF-β in promoting fibrosis [182].

In a clinical study from Ishikawa, losartan reduced PAI1 in renal allograft recipients treated with cyclosporine [183]. Among the 12 participants recruited for the study, four kidney transplant recipients were treated with losartan, while the rest served as controls. PAI1 levels at the end of two years of treatment were lower. In a follow-up publication, they performed a microscopic evaluation of renal biopsy cores and found decreased interstitial fibrosis 2 years after study initiation among participants who were administered losartan when compared to transplant patients not receiving an ARB.

Animal and cell culture studies are exploring the mechanisms underlying effects of losartan on kidney fibrosis. Huang et al. found that losartan can hinder signal mechanotransduction involved in EMT of renal epithelial cells. Losartan treatment reduced fibrin deposition and luminal vacuolization in mice with renal fibrosis induced via unilateral ureteral obstruction. The drug protects against biomechanical stresses that promote myofibroblast formation and deposition during EMT [184].

### 5.2. Mineralocorticoid Receptor Antagonists

Aldosterone, the final signaling hormone of the RAAS, is a mineralocorticoid that promotes sodium reabsorption and potassium excretion in the collecting ducts, regulating blood pressure and electrolyte balance [185]. The hormone also promotes proinflammatory activity that can contribute to the fibrotic damage of organs such as the heart and kidney. MR activation promotes the creation of ROS such as superoxide and hydrogen peroxide via NADPH oxidase [186,187]. In the kidney, aldosterone can promote inflammation and fibrosis through the activation of the MR found in various cellular compartments such as the podocytes, mesangial cells, epithelial cells, and myeloid cells [188]. MR activation in mesangial cells and podocytes stimulates transcription of various inflammatory mediators such as PAI1, TGF-β, nuclear factor (NF)-kB, and IL-6 that contribute to glomerular damage [189,190]. In animal models of kidney disease, aldosterone promotes the growth of renal fibroblasts via the activation of growth factor receptors and can directly encourage the production of profibrotic cytokines and matrix proteins [191]. In murine models of diabetic kidney disease, MR antagonists have been shown to reduce kidney fibrosis [192,193].

MR antagonists may be steroidal or non-steroidal [194]. Steroidal MR antagonists such as spironolactone have a lower safety profile due to high incidences of hyperkalemia in patients with CKD, and this is a persistent hindrance in trials [195]. Non-steroidal MR antagonists (i.e., apararenone, esaxernone, finerenone) have a better safety profile, and in clinical trials they avoid complications associated with the steroidal MR antagonists, without losing drug efficacy [196,197]. Clinical studies demonstrate slowing of progression with the use of non-steroidal MR antagonists in diabetic CKD [198]. The FIDELIO-DKD (FInerenone in reducing kiDnEy faiLure and dIsease prOgression in Diabetic Kidney Disease) trial randomized patients with type 2 diabetes and compromised kidney function to receive oral finerenone (10 mg or 20 mg) or placebo over a median follow-up interval of 2.6 years [199]. Although kidney fibrosis was not the primary outcome measured in the FIDELIO-DKD trial, the trial demonstrated that finerenone improved kidney outcomes in patients with diabetic kidney disease and reduced the risk of fatal secondary outcome events. [200,201]. The FIGARO-DKD trial produced similar results to the FIDELIO-DKD clinical trial, confirming the ability of finerenone to reduce proteinuria, albuminuria, and slow the decline in eGFR in CKD patients. Data from both key clinical trials were combined under the FIDELITY analysis, which provided even greater evidence supporting the efficacy and safety of finerernone in patients with CKD and type 2 diabetes mellitus. The drug demonstrated both cardiovascular and renal benefits, and specifically reduced the risk of kidney failure as suggested by a persistent decrease in estimated GFR or the need for a kidney transplant [202].

Data on the effect of steroidal MR antagonists in patients with CKD are limited [203]. A Cochrane review demonstrated that in patients with CKD, spironolactone reduced proteinuria and systolic/diastolic blood pressure [204].

### 5.3. Sodium/Glucose Cotransporter 2 Inhibitors

SGLT2i (gliflozins) are a class of oral hypoglycemic agents that improve renal and cardiovascular outcomes and mortality in patients with type 2 diabetes [205,206]. They also slow the decline in GFR in persons with CKD without diabetes [207]. The main mechanism of action of these drugs involves direct inhibition of glucose reuptake by the kidney via SGLT2 transporters located in the proximal tubules. This leads to enhanced glucose excretion in the urine, reduced blood glucose concentration, and lowered HbA1c [205,208]. These drugs promote weight loss, increase insulin sensitivity and lipid metabolism, and decrease lipotoxicity [209]. By fostering sodium excretion, gliflozins reduce intraglomerular capillary pressure and are key nephroprotective drugs [210,211]. SGLT2i also seem to control extracellular matrix deposition by hindering the epithelial-to-mesenchymal transition that promotes renal fibrosis over time. Guo et al. demonstrated that in streptozotocin-induced models of diabetic nephropathy, mice treated with dapagliflozin displayed remission of pathological lesions such as glomerular sclerosis, thickening of the glomerular basement membrane (GBM), podocyte injury in the glomeruli, decreased nephrotoxic levels, and decreased SGLT2 expression. Elisa evaluation demonstrated decreased circulating levels of insulin-like growth factor-1 (IGF1) and insulin-like growth factor-2 (IGF2) in mice treated with dapagliflozin. By regulating the insulin-like growth factor-1 receptor (IGF1R)/phosphoinositide 3-kinase (PI3K) regulatory axis, the drug reduced glomerular injury by regulating podocyte EMT [212]. Experimental findings also indicate that gliflozins reduce inflammatory markers such as IL-6, TNF, interferon (IFN)-γ, NF-κβ, toll-like receptor (TLR)-4, and TGF-β; improve mitochondrial function; and reduce mesangial and myofibroblast growth [210,213]. In a transcriptome analysis from Pirklbauer et al., the drug empagliflozin demonstrated anti-inflammatory effects in human proximal tubular cells by inhibition of IL-1β inflammatory pathway gene expression [214]. Although the exact mechanism behind the antifibrotic effects of gliflozins is indeterminate, these drugs protect kidney podocytes from the detrimental effects of hyperfiltration, albuminuria-induced tubular injury, and glucotoxicity. The anti-inflammatory and antifibrotic effects of this drug class may be secondary to the metabolic and hemodynamic alterations of SGLT2 inhibition [215].

The benefits of SGLT2 inhibitors in patients with non-diabetic-related kidney disease remain unclear. Preclinical studies assessing the effectiveness of the drug in nondiabetic animal models of CKD have given contradictory results. Some studies found no renoprotective effects with the use of the drug, while others demonstrated a reduction in the number of glomerular lesions, podocyte damage, proteinuria, renal hyperfiltration, microalbuminuria, and markers for renal inflammation [216,217]. Differences in animal models of CKD used make it difficult to compare results. Clinical trials assessing the effectiveness of SGLT2 in patients with non-diabetic CKD are also limited. Rajasekeran et al. studied the impact of dapagliflozin as an adjunct to RAAS blockade in patients with focal segmental glomerulosclerosis and found that dapagliflozin yielded no additional renal hemodynamic or antiproteinuric effects in patients [218]. Similarly, Greeviroj et al. studied the use of canagliflozin on non-diabetic obese individuals, and participants demonstrated a reduction in body weight but no significant renal benefits [219]. Nevertheless, post hoc analysis of results from the CREDENCE trial (which studied the effects of canagliflozin on renal and cardiovascular outcomes in patients with diabetic nephropathy) and several other cardiovascular outcome trials using SGLT2 inhibitors have demonstrated benefits independent of their glucose-lowering capabilities [220,221]. The potential of this drug class to produce renoprotective effects in patients with non-diabetic CKD remains in question until more definitive future trials are conducted [222].

## 6. Antifibrotic Drugs in Clinical Trials

### 6.1. Pirfenidone

Pirfenidone is an antifibrotic drug whose mechanism of action is thought to involve inhibition of TGF-β [223]. In preclinical studies, pirfenidone blocked the expression of TGF-β and its subsequent downstream effects [224]. In cell culture experiments, the drug was able to inhibit the production and signaling of TGF-β as well as the downstream production of reactive oxygen species (ROS) [225]. Anti-inflammatory effects are exerted through the decreased expression of tumor necrosis factor (TNF)-α, interleukin (IL)-1, and IL-6 [226,227,228]. In mouse models of diabetic kidney disease, animals that were administered the drug displayed reduced mesangial matrix expansion without affecting albuminuria [229]. Pirfenidone also acts via inhibition of mitogen-activated protein kinase (MAPK) signaling [230,231]. MAPK and TGF-β pathways interact with each other and work synergistically to promote fibrosis [232,233,234]. MAPK pathway inhibition attenuates TGF-β-induced collagen overexpression in fibroblasts and mesangial cells [235,236].

Other preclinical animal studies have demonstrated the efficacy of the drug in kidney fibrosis of varying etiologies. In rats that have undergone partial nephrectomy, pirfenidone reduced collagen accumulation in the remaining organ [237]. The drug reduced chronic cyclosporine-induced tubulointerstitial fibrosis and relieved intratubular fibrosis in rats with unilateral ureteral obstruction [227,238,239,240].

Pirfenidone is approved by the US Food and Drug Administration (FDA) in the treatment of idiopathic pulmonary fibrosis (IPF) and is excreted predominantly by the kidneys. The drug will accumulate in patients with reduced renal function; hence, the FDA cautions against the use of the drug in patients with an eGFR less than 80mL/min. In an open-label pilot study, pirfenidone demonstrated a desirable effect in slowing the loss of GFR in patients with focal sclerosing glomerulonephritis (FSGS) and a mean baseline estimated GFR of 26 ± 9.4 mL/min per 1.73 m^2^. However, there were limitations to this study that included the absence of a placebo control and the selection of GFR rate decline as the primary outcome variable. Determining the rate of GFR decline via multiple serum creatinine values can be an issue since a single outlier can obscure the overall slope and interfere with the accuracy of representation of the effect of the drug on this endpoint [241,242]. The TOP-CKD is a phase 2 clinical trial that consists of 200 individuals with CKD with an eGFR greater than 20 mL/min and a 1% risk progression to end-stage renal disease (ESRD) over the next 5 years [227,243,244]. The trial is a randomized, double-blind, placebo-controlled study in which participants receive either pirfenidone or placebo treatment for 12 months, followed by a 6-month no-treatment follow-up period. The study utilizes noninvasive methods such as diffusion MRI scans and urinary biomarkers of renal fibrosis to evaluate if pirfenidone can serve as a promising new antifibrotic treatment in patients with CKD. The trial is still ongoing and is estimated to be completed by December 2024.

### 6.2. Silencing of MicroRNA-21 and Lademirsen

Lademirsen is an antisense oligonucleotide designed specifically to inhibit microRNA (miR)-21, a small noncoding RNA that is a known driver of kidney fibrosis [245,246]. Unfortunately, the drug failed in human studies. It was investigated in a phase 2 double-blind, placebo-controlled clinical trial for Alport Syndrome. The study, which was set to be completed by June 2023, aimed to assess safety and change in GFR from baseline to 48 weeks in patients whose eGFR was between 36–89 mL/min and who were at risk for rapid progression of renal decline despite RAS blockade. Although the primary endpoints did not address kidney fibrosis, blood and urine TGF-β1 were analyzed. The trial was terminated early because interim analysis did not demonstrate significant improvement in kidney function with lademirsen compared to placebo [247,248].

The results were disappointing in light of the literature supporting therapeutic targeting of miR-21, the expression of which is stimulated via the TGF-β/Smad3 signaling pathway in renal tubular epithelial cells [249]. MiRNA-21 promotes fibrosis in various organs by silencing metabolic pathways that are crucial for ATP production, increasing the production of ROS, and inducing inflammatory signaling pathways [250]. In studies involving murine models of Alport nephropathy, miR-21 downregulation improved survival by reducing glomerulosclerosis, interstitial fibrosis, tubular injury, and inflammation [251,252]. Development of lademirsen has been halted, but the concept of miRNA manipulation is being carried forward, as discussed in the following section on future treatments [248,253].

## 7. Future Treatments and Potential Approaches

### 7.1. Strategies Targeting TGF-β

At this time, renal interstitial fibrosis therapy directed at regulating TGF-β or TGF-β/Smad signaling is still lacking. As diabetic kidney disease and CKD continue to become more prevalent across the globe, it is becoming increasingly important to identify new treatments, and there are several up-and-coming approaches to therapy specifically aimed at inhibiting this pathway [254,255]. Based on animal studies described in the previous section, LRG1 is considered a promising TGF-β-targeting therapy in diabetes-related and non-diabetes-associated renal fibrosis [128,256].

The ECM protein microfiber-associated protein 4 (MFAP4) is also being studied as a contributor to kidney fibrosis [257,258]. MFAP4 knockout mice exhibit reduced activation of the TGF-β/Smad pathway compared to wild-type mice after unilateral ureteral obstruction, leading to less renal fibrosis in the MFAP4-deficient state [257]. Additionally, Klotho-derived peptide (KP)1 has been shown to provide protection against TGF-β/Smad-driven renal fibrosis in mice [88,259,260]. KP1 works by blocking TGF-β-induced activation of Smad2/3, and thus reduces renal fibrosis and damage [88]. Small-molecule inhibitors of TGF-β1 are also under evaluation for their anti-fibrotic properties [261,262].

### 7.2. RNA-Based Therapeutics

Another potential therapeutic approach to renal fibrosis is the use of small interfering RNAs (siRNA). These molecules can markedly reduce expression of disease-related proteins. Liu et al. experimented with siRNA targeted to the kidney and directed against TGF-β1 [263]. They created nanoparticles comprised of cationic liposomes containing TGF-β1-siRNA and coated with non-inhibitory plasminogen activator inhibitor 1R (PAI-1R), a ligand that targets glomerular cells. When they injected these nanoparticles into a nephritic rat model, they honed in on the glomeruli and were able to silence TGF-β1 mRNA and subsequent protein expression specifically in the glomeruli, leading to reduced glomerular matrix accumulation. Thus, utilizing siRNA in the treatment of renal fibrosis seems to be a promising option to be explored further.

MicroRNAs (miRNAs), single-stranded noncoding RNAs that can regulate target gene expression either by blocking mRNA translation or promoting mRNA degradation, impact expression of many kidney-related fibrotic proteins. MiR-26a specifically exerts antifibrotic effects and is downregulated in certain fibrotic diseases of the kidney, heart, and lungs [264]. Zheng et al. induced tubulointerstitial fibrosis in mice using aldosterone and found increased CTGF in the kidney tissue [265]. They then treated the mice with exosomes (small microvesicles useful as an RNA delivery system) enriched in miR-26a and found that this treatment reduced fibrosis. The exosomes protected the miR-26a from degradation while conveying it to the kidney after injection into the caudal vein. The kidneys of mice treated with miR-26a exosomes had lower expression of CTGF than those treated with control exosomes. In companion cell culture studies, aldosterone-treated mouse tubular epithelial cells exposed to exosomes carrying miR-26a had reduced EMT and inhibition of the CTGF/SMAD signaling pathway.

Another potential RNA-based treatment for renal fibrosis is through the use of relaxin-2 mRNA. Relaxin-2 is an anti-fibrotic molecule in kidney cells being studied as a target for upregulation as a means to inhibit the development of renal fibrosis [266]. This molecule inhibits downstream signaling of TGF-β1 and has previously been shown to decrease the deposition of collagen in experimental murine models of both acute and chronic kidney disease [267,268]. Ding et al. utilized crystalline nanoparticles known as cubosomes to deliver relaxin-2 mRNA to the kidney in mice with unilateral ureter obstruction [269]. In this model, it was demonstrated that cubosomes loaded with relaxin-2 mRNA reduced kidney injury by decreasing fibrotic and inflammatory responses.

### 7.3. Utilizing Transcriptional Regulators Snail1 and Twist1 as Therapeutic Targets

Specific pathways and mediators can be manipulated to limit renal fibrosis and CKD as a whole. Two transcriptional regulators of EMT are Twist family basic helix-loop-helix (BHLH) transcription factor (Twist)1 and Snail family transcriptional repressor (Snail)1 [270,271,272]. Lovisa et al. found that when subjected to treatments that induced renal fibrosis, mice with conditional deletion of either of these two transcription factors in their proximal renal tubular epithelial cells had reduced EMT and better renal function with less fibrosis as compared to wild-type mice [273]. As further evidence that Snail1 induces EMT, Grande et al. found that unilateral ureteral obstruction reactivates Snail1 in mouse renal epithelial cells, and this reactivation is a requirement for renal fibrosis to occur [270]. In the same study, the researchers tested not only whether renal fibrosis could be prevented by keeping Snail1 inactive, but also whether it could be reversed by blocking Snail1 expression. They reduced expression by targeting a splicing site in the Snail1 mRNA, and this attenuated fibrosis in vivo. The treated mice had lower collagen deposition and had better morphology as compared to mice in which Snail1 expression was intact [274].

Furthermore, Qi et al. showed that Snail1-induced partial EMT leads to cell cycle arrest and p53–p21 axis upregulation in both human kidney allografts with interstitial fibrosis and murine models [275]. Expression of p53 and p21 in RTEC were significantly increased in mice with unilateral ureteral obstruction that were injected with a plasmid encoding Snail1 mRNA. These two tumor suppressor genes arrest the RTEC in the G2/M phase, leading to an increase in the release of inflammatory cytokines, notably NF-κB. In mice with unilateral ureteral obstruction and overexpression of Snail1, the partial EMT process and the p53–p21-mediated cell cycle arrest were mitigated by NF-κB pathway inhibition. Taken together, the study results led to the conclusion that there is a reciprocal positive loop between partial EMT and G2/M arrest of RTEC, and NF-κB-mediated inflammation is likely the underlying mechanism. Targeting of NF-κB could interrupt the loop and be reparative with a beneficial decrease in or reversal of renal fibrosis.

Further examination of anti-fibrotic agents aimed at Snail1 and Twist1 may yield treatments for reducing EMT and treating renal fibrosis.

### 7.4. Mesenchymal Stem Cells

There have been numerous studies that have demonstrated the effectiveness of mesenchymal stem cells (MSC) in reducing organ fibrosis in the liver, lung, heart, and kidney [276,277,278]. MSC are derived from various sources including bone marrow, adipose tissue, umbilical cord, amniotic membrane, chorionic membrane, placenta decidua, and Wharton colloid [279,280,281]. They have the capacity to differentiate into numerous cell lineages like osteoblasts, adipocytes, and chondrocytes in vitro [282]. Their pro-angiogenic, immunosuppressive, and anti-fibrotic abilities enable them to combat inflammation and promote tissue repair, making them a cell population potentially capable of treating fibrosis [276,283]. Depending on the source from which these cells are derived, MSC serve as protective fibrotic mediators by hindering specific phases of the renal fibrotic process [284,285].

Bone marrow-derived MSC inhibit the cellular activation and inflammatory injury phase of CKD by inhibiting the expression of proinflammatory cytokines. Bone marrow-derived MSC also hinder profibrogenic signaling pathways such as TGF-β1/Smad, NK-κB, and ERK [286]. They restrain the epithelial-to-mesenchymal transition phase that promotes ECM deposition. Preclinical studies with bone marrow-derived MSC corroborate that these cell lines can improve kidney function in animal models of CKD. In mouse models of CKD, transplanted bone marrow-derived MSC reduced blood urea nitrogen (BUN) and urine albumin to creatinine ratio. There was also a reduction in mRNA transcripts of renal fibrosis-related indicators like collagen IV, fibronectin, and α-SMA [287,288].

Bone marrow-derived MSC are the most commonly used cell derivatives for CKD, and their reduced expression of MHC I and MHC II make them less likely to be attacked by allogeneic T cells compared to other MSC derivatives. However, bone marrow-derived MSC are harder to purify quickly and efficiently than other types of MSC for kidney regeneration [289].

Research aimed at enhancing the therapeutic and antifibrotic capabilities of MSC via preconditioning with various methods has been explored. Utilizing a serum-free medium has been shown to protect MSC from senescence-like age-related deterioration characteristics and preserve their spindle shape and morphology during culture [290].

MSC cultured using serum-free medium that had also undergone hypoxic preconditioning exhibited synergistic enhancement of their proliferative and migratory capacities. Further, exposure of TGF-β1-stimulated human renal tubular epithelial cells (HK-2) to MSC that had received combined hypoxic preconditioning and serum-free medium demonstrated better inhibition of the TGF-β/Smad signaling pathway compared to HK-2 exposed to MSC treated only with serum-free medium. Moreover, in an in vivo rat model of kidney fibrosis brought on by ischemia-reperfusion injury, rats given MSC that had been prepared with a combination of hypoxic preconditioning and serum-free media showed greater mitigation of renal fibrosis by the MSC than rats given MSC treated with serum-free medium alone.

Umbilical cord-derived MSC, specifically the stem cells harvested from Wharton’s Jelly, demonstrate promise for future clinical applications because of their limited heterogeneity and easy accessibility in numerous tissues [291,292]. Compared to bone marrow-derived MSC, umbilical cord-derived MSC have lower immunogenicity and superior ability to proliferate and differentiate, and they can be isolated without any invasive surgical procedure and manufactured in large quantities without compromising potency [289,293,294]. Wharton’s jelly-derived MSC can be obtained easily from what is considered medical waste, avoiding ethical controversies that arise from stem cell use [291,295].

Studies have also assessed the use of MSC as an adjunct to current pharmacological approaches of RAAS blockade for CKD progression. Maires et al. used adipose tissue-derived MSC in conjunction with the angiotensin receptor-blocking agent losartan to investigate synergistic effects on CKD progression in rats and found co-treatment with both agents resulted in significant improvement and regression of parameters such as proteinuria and albuminuria [296]. Compared to either monotherapy, dual treatment was also associated with regression of structural glomerular injury and better preservation of glomerular proteins. Likewise, hypertension and interstitial macrophage infiltration were more effectively reduced compared to either treatment alone.

Although there are a number of CKD therapies employed currently and some promising approaches in development (Table 1), we are not yet able to halt the progression of fibrosis.

### 7.5. Src Family Kinases

An alternative therapeutic approach for CKD is via Src Family Kinases (SFKs). These non-receptor tyrosine kinases play a pivotal role in the regulation of cell growth and differentiation under normal physiologic conditions [297]. However, under pathological conditions, SFKs are implicated in tumor cell adhesion, migration, and metastasis [298]. Multiple studies have demonstrated the effectiveness of SRK inhibition in decreasing tissue fibrosis, specifically in the lung, pancreas, and skin [299,300,301]. Li et al. highlighted the significance of the Lyn family of SFKs. Lyn is of particular importance because its deficiency has led to the pathogenesis of many different diseases [302]. For instance, in mice, Lyn knockout exacerbates lipopolysaccharide-induced lung inflammation, and the mechanism involves NF-κB activation [303]. Furthermore, in relation to the kidneys, mice lacking Lyn exhibit severe kidney disease, including immune complex-mediated lupus nephritis [304]. Nintedanib, a tyrosine kinase inhibitor that inhibits phosphorylation of several SFKs, effectively reduces kidney fibrosis and inflammation in mouse models [305,306,307].

Fyn, also a member of the Src family, may protect against diabetic kidney injury. Knockdown of Fyn by siRNA in a murine renal proximal tubular epithelial cell line under high glucose conditions abrogated the glucose-driven upregulation of the stress proteins metallothionein-1/2 [308,309]. Metallothionein-1/2 induction acts as a protective antioxidant mechanism; therefore, support of this mechanism by Fyn may be a therapeutic avenue [310].

Inhibiting phosphorylation of SFKs lessens the activation of renal interstitial fibroblasts, reduces the deposition of ECM in the kidney, and ultimately attenuates fibrosis. In the rat renal interstitial fibroblast NRK-49F cell line, exposure to an Src tyrosine kinase inhibitor has been shown to reduce activation, proliferation, and TGF-β signaling [311]. Hematopoietic cell kinase (HCK), another SFK family member, has been studied in renal biopsy specimens from kidney transplant patients at 12 months post-transplant. In these specimens, HCK was observed to co-localize with macrophages, and high HCK levels were associated with inflammation and fibrosis. Further affirming a role for HCK as a driver of kidney fibrosis, HCK knockout attenuated fibrosis in a mouse model of unilateral ureteral obstruction [312].

Although effective in preclinical models, nephrotoxicity and case reports of renal thrombosis with tyrosine kinase inhibitors make clinical translation to humans difficult [313,314].

## 8. Conclusions

CKD is a serious health condition associated with loss of kidney function over time. CKD has a broad range of effects that encompass many systems including the hematologic, cardiovascular, nervous, and endocrine systems. The high rate of morbidity and mortality in affected patients underscores the critical need for ongoing research targeted at developing novel medical treatments. Kidney fibrosis, a major pathophysiologic change seen in CKD that impedes renal blood supply and function, is a key focus in the search for new therapeutic options. TGF-β, the critical molecule associated with renal fibrosis, can be utilized to detect the severity of fibrosis and is a target for drug treatment. Although clinical treatment for CKD has a long way to go, utilizing RNA-based therapeutics, transcriptional regulators, and stem cells are promising approaches.

## Figures and Tables

**Figure 1 jcm-13-01881-f001:**
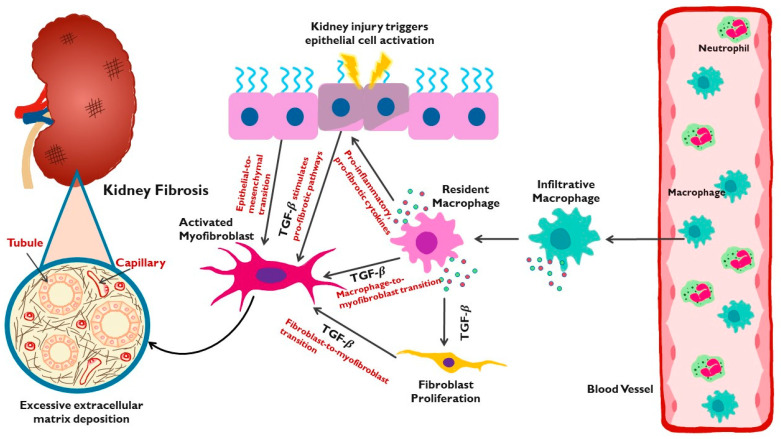
Schematic of the key elements in the process of kidney fibrosis. Injury to the kidney initiates activation of epithelium, proliferation of fibroblasts, and inflammation with infiltration of macrophages. Transforming growth factor- β (TGF-β) and other cytokines are released by kidney cells, stimulating transdifferentiation of epithelial cells to myofibroblasts, macrophages to myofibroblasts, and fibroblasts to myofibroblasts. TGF-β activates a cascade of pro-fibrotic downstream signaling pathways. Myofibroblasts are the cell type primarily responsible for production of excessive extracellular matrix, which leads to scarring, structural damage, and kidney fibrosis.

**Table 1 jcm-13-01881-t001:** Treatment approaches for kidney fibrosis.

Current Status	Treatment	Description	References
In clinical use	Renin-angiotensin blockers	Inhibit the RAAS, slowing CKD progression and kidney fibrosis; also reduces the activity of TGF-β, PAI1, and PDGF, all molecules involved in promoting glomerular damage.	[168,169,170,171,172,183]
In clinical use	Mineralocorticoid receptor (MR) antagonists	Inhibit the activation of aldosterone at the MR, reducing inflammation and proteinuria; block the transcription of various inflammatory mediators that contribute to kidney fibrosis such as PAI1, TGF-β, NF-kB, and IL-6.	[194,195,196,197,198,199,200,201,202,203,204]
In clinical use	Sodium-glucose cotransporter-2 (SGLT-2) inhibitors	Inhibit glucose reuptake by the kidney via SGLT-2 transporters located in the proximal tubules; promote lower macrophage-mediated inflammation and cytokines such as IL-6, TNF-α, IFNγ, NF-κβ, TLR-4, and TGF-β, thus reducing glomerular fibrosis. They also lower glomerular pressure and improve renal hemodynamics. Protective against ESRD even in patients without type 2 diabetes.	[205,206,207,208,209,210,211,219,220,221,222]
Experimental	Pirfenidone	Antifibrotic orally administered drug; works mainly via inhibition of TGF-β and MAPK signalling. Has anti-inflammatory and antioxidant activity.	[223,224,225,230,231,232,233,234,235,236]
Experimental	MicroRNA silencing	Antisense oligonucleotides that silence specific fibrosis-related microRNAs are being explored. Lademirsen, an anti-microRNA-21 drug, failed in human studies, even though microRNA21 is deregulated in kidney fibrosis. Antisense oligonucleotides that silence other microRNAs may be a viable treatment.	[245,246,247,248,249,250,251,252,253]
Experimental	Mesenchymal stem cells (MSC)	Multi-potent adult stem cells that demonstrate anti-fibrotic and anti-inflammatory abilities and can induce repair and regeneration.	[272,279]

Abbreviations. Chronic kidney disease (CKD); end-stage renal disease (ESRD); interleukin-6 (IL-6); interferon-γ (IFN-γ); mitogen-activated protein kinase (MAPK); mesenchymal stem cells (MSC); mineralocorticoid receptor (MR); nuclear factor-kB (NF-kB); plasminogen activator inhibitor-1 (PAI1); platelet-derived growth factor (PDGF); renin-angiotensin-aldosterone system (RAAS); sodium-glucose cotransporter-2 (SGLT-2); toll-like receptor-4 (TLR-4); transforming growth factor- β (TGF-β); tumor necrosis factor-α (TNF-α).

## Data Availability

Not applicable.

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
