# Peer review of "Fibrosis in Chronic Kidney Disease: Pathophysiology and Therapeutic Targets"

_jcm, 2024, doi:10.3390/jcm13071881_

Round 1

Reviewer 1 Report

Comments and Suggestions for Authors

I have reviewed the article entitled, 'Fibrosis in Chronic Kidney Disease: Pathophysiology and Therapeutic Targets'.

Authors have focused on the current therapies and future directions for the treatment of Chronic Kidney Disease. 

There are some comments/suggestions-

1. In the article, authors should use heading or subheadings with full form, instead of short-forms.

2. As it is a review article, author should not mention a heading "results". No

Use a proper heading to describe it.

3. Author has done a detailed review of the reports and have cited 292 references. There are many relevant articles cited which should be removed.

4. In table 1, Author should include one column for references.

If the data has been mentioned in the table, do not repeat it in the text.

5. Author should use the firmed statement about the published reports. Unlike line no. 478, 'action is thought to involve'... It is an example. There are so many statements written in the similar way.

6. Authors should recheck the review article again to make it more crisp and presentable.

7. The title of the article could be improved.

Comments on the Quality of English Language

Grammar should be checked with a professional software.

Author Response

We thank the reviewer for thoroughly scrutinizing our manuscript. As requested, we have revised the manuscript and addressed the specific comments of the reviewer. The revised sections are delineated in red in a marked copy of the manuscript text.

Below, we provide a point-by-point response to the reviewer’s comments.

Reviewer # 1 Comments

  • COMMENT #1: In the article, authors should use heading or subheadings with full form, instead of short-forms.

RESPONSE: We have changed the headings as suggested.

  • COMMENT #2: As it is a review article, author should not mention a heading "results". No. Use a proper heading to describe it.

     RESPONSE: We have removed “results”.

  • COMMENT #3: Author has done a detailed review of the reports and have cited 292 references. There are many relevant articles cited which should be removed.

RESPONSE: We have written many review articles and generally include this number of references so that our paper can serve as a handy resource for delving deeper into specific areas. Also, reviewers generally want us to add more information and that necessitates additional references as is the case here. We have eliminated a few references, but then added more as we responded to reviewer requests and added 2 new subsections.

  • COMMENT #4: In table 1, Author should include one column for references. If the data has been mentioned in the table, do not repeat it in the text.

RESPONSE: We have modified the table and now include a column for references as suggested. We mention the data in both places as this is customary in general in reviews. The table serves as a summary, but the topic is discussed in the text.

  • COMMENT #5: Author should use the firmed statement about the published reports. Unlike line no. 478, 'action is thought to involve'... It is an example. There are so many statements written in the similar way.

     RESPONSE: We have changed this as suggested and where appropriate.

  • COMMENT #6: Authors should recheck the review article again to make it more crisp and presentable..

     RESPONSE: We have rechecked and made it more crisp.

  • COMMENT #7: The title of the article could be improved.

     RESPONSE: The title perfectly fits the content.

We thank the reviewer and believe that the manuscript is improved as a result of their input.  We hope you will agree, and decide in favor of accepting our report at this time.

Reviewer 2 Report

Comments and Suggestions for Authors

This review provides detailed information about the characteristics of kidney fibrosis, assessment, mechanism, CKD and fibrosis drugs in use, and drugs in clinical trials. I only have some comments and suggestions that may help with improvement.

1.      The segment on kidney fibrosis offers a foundational understanding but lacks critical details on the fibrinogenic niche and the composition of extracellular matrix (ECM) proteins. Incorporating this information would provide readers with a comprehensive definition and components of kidney fibrosis, establishing a clearer conceptual framework right from the outset.

2.      While the review effectively summarizes the roles of TGF-β signaling, diabetic nephropathy (DN), epithelial-mesenchymal transition (EMT), and serum amyloid A (SAA), it overlooks significant evidence on the involvement of Src family kinases in CKD, kidney inflammation, and fibrosis. Several studies, as indicated by references [PMID: 28615246, 35883540], highlight the pivotal role these kinases play. It is recommended that the review be augmented with insights into this area to encompass a broader spectrum of mechanistic pathways involved in kidney fibrosis.

3.       The schematic figure presented in the review appears overly simplistic and perhaps not at the level of sophistication expected in a scientific publication. It would be beneficial to consult and emulate the style of figures found in publications such as Nature Reviews Nephrology. Redesigning the figure with a higher degree of scientific accuracy and visual maturity could significantly elevate the review's overall quality and reader engagement.

Author Response

RE:  Manuscript ID: jcm-2895521 Review Article “Fibrosis in Chronic Kidney Disease: Pathophysiology and Therapeutic Targets”

We thank the reviewer for thoroughly scrutinizing our manuscript. As requested, we have revised the manuscript and addressed the specific comments of the reviewer. The revised sections are delineated in red in a marked copy of the manuscript text.

Below, we provide a point-by-point response to the reviewer’s comments.

Reviewer # 2 Comments

  • COMMENT #1: The segment on kidney fibrosis offers a foundational understanding but lacks critical details on the fibrinogenic niche and the composition of extracellular matrix (ECM) proteins. Incorporating this information would provide readers with a comprehensive definition and components of kidney fibrosis, establishing a clearer conceptual framework right from the outset.

RESPONSE: We have elaborated on this topic in a new subsection (4.1. The Fibrotic Niche and The Profibrotic Microenvironment) with additional references.

  • COMMENT #2: While the review effectively summarizes the roles of TGF-β signaling, diabetic nephropathy (DN), epithelial-mesenchymal transition (EMT), and serum amyloid A (SAA), it overlooks significant evidence on the involvement of Src family kinases in CKD, kidney inflammation, and fibrosis. Several studies, as indicated by references [PMID: 28615246, 35883540], highlight the pivotal role these kinases play. It is recommended that the review be augmented with insights into this area to encompass a broader spectrum of mechanistic pathways involved in kidney fibrosis.

RESPONSE: We have added the material in a new subsection (7.5. Src Family Kinases) with references as suggested.

  • COMMENT #3: The schematic figure presented in the review appears overly simplistic and perhaps not at the level of sophistication expected in a scientific publication. It would be beneficial to consult and emulate the style of figures found in publications such as Nature Reviews Nephrology. Redesigning the figure with a higher degree of scientific accuracy and visual maturity could significantly elevate the review's overall quality and reader engagement..

RESPONSE: We have redesigned and refined the figure.

We thank the reviewer and believe that the manuscript is improved as a result of their input.  We hope you will agree, and decide in favor of accepting our report at this time.

Round 2

Reviewer 1 Report

Comments and Suggestions for Authors

The author has answered to the comments.

Author Response

We thank the reviewer for thoroughly scrutinizing our manuscript.

Below, we provide a point-by-point response to the reviewer’s comments.

Reviewer # 1 Comments

  • COMMENT #1: The author has answered to the comments.

            RESPONSE: We thank the reviewer and appreciated the helpful suggestions.

Reviewer 2 Report

Comments and Suggestions for Authors

The revised review added some information on SRC kinases in kidney fibrosis, but still missed important references for HCK, like PMID: 37463911, and SRC, like PMID: 26444028, PMID: 35918533. They should add these important references. 

Author Response

We thank the reviewer for thoroughly scrutinizing our manuscript. As requested, we have revised the manuscript and addressed the specific comment of the reviewer. The revised sections are delineated in red in a marked copy of the manuscript text.

Below, we provide a point-by-point response to the reviewer’s comments.

Reviewer # 2 Comments

  • COMMENT #1: The revised review added some information on SRC kinases in kidney fibrosis, but still missed important references for HCK, like PMID: 37463911, and SRC, like PMID: 26444028, PMID: 35918533. They should add these important references.

RESPONSE: We have elaborated further on this topic with the additional references indicated. PMID: 37463911 is Reference 312, PMID: 26444028 is Reference 311, and PMID: 35918533 is Reference 308.

We thank the reviewer and believe that the manuscript is improved as a result of their input.  We hope you will agree, and decide in favor of accepting our report at this time.
